# Learning to Incentivize Other Learning Agents

**Jiachen Yang**[1,*] **Ang Li**[2], **Mehrdad Farajtabar**[2], **Peter Sunehag**[2], **Edward Hughes**[2], **Hongyuan Zha**[1,3,†]

[1]Georgia Institute of Technology    [2]DeepMind
[3]AIRS and Chinese University of Hong Kong, Shenzhen
jiachen.yang@gatech.edu
[anglili,farajtabar,sunehag,edwardhughes]@google.com
zha@cc.gatech.edu

## Abstract

The challenge of developing powerful and general Reinforcement Learning (RL) agents has received increasing attention in recent years. Much of this effort has focused on the single-agent setting, in which an agent maximizes a predefined extrinsic reward function. However, a long-term question inevitably arises: how will such independent agents cooperate when they are continually learning and acting in a shared multi-agent environment? Observing that humans often provide incentives to influence others' behavior, we propose to equip each RL agent in a multi-agent environment with the ability to give rewards directly to other agents, using a learned incentive function. Each agent learns its own incentive function by explicitly accounting for its impact on the learning of recipients and, through them, the impact on its own extrinsic objective. We demonstrate in experiments that such agents significantly outperform standard RL and opponent-shaping agents in challenging general-sum Markov games, often by finding a near-optimal division of labor. Our work points toward more opportunities and challenges along the path to ensure the common good in a multi-agent future.

## 1   Introduction

Reinforcement Learning (RL) [37] agents are achieving increasing success on an expanding set of tasks [28, 20, 32, 41, 6]. While much effort is devoted to single-agent environments and fully-cooperative games, there is a possible future in which large numbers of RL agents with imperfectly-aligned objectives must interact and continually learn in a shared multi-agent environment. The option of centralized training with a global reward [13, 35, 31] is excluded as it does not scale easily to large populations and may not be adopted by self-interested parties. On the other hand, the paradigm of decentralized training—in which no agent is designed with an objective to maximize collective performance and each agent optimizes its own set of policy parameters—poses difficulties for agents to attain high individual and collective return [29]. In particular, agents in many real world situations with mixed motives, such as settings with nonexcludable and subtractive common-pool resources, may face a social dilemma wherein mutual selfish behavior leads to low individual and total utility, due to fear of being exploited or greed to exploit others [30, 23, 24]. Whether, and how, independent learning and acting agents can cooperate while optimizing their own objectives is an open question.

The conundrum of attaining multi-agent cooperation with decentralized training of agents, who may have misaligned individual objectives, requires us to go beyond the restrictive mindset that the collection of predefined individual rewards cannot be changed by the agents themselves. We draw inspiration from the observation that this fundamental multi-agent problem arises at multiple

---

[*]Work done during internship at DeepMind
[†]On leave from College of Computing, Georgia Institute of Technology

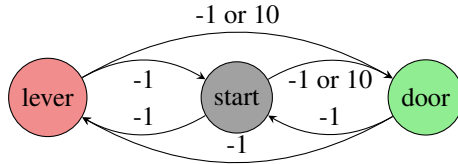

Figure 1: The $N$-player *Escape Room* game ER$(N, M)$. For $M < N$, if fewer than $M$ agents pull the lever, which incurs a cost of $-1$, then all agents receive $-1$ for changing positions. Otherwise, the agent(s) who is not pulling the lever can get $+10$ at the door and end the episode.

scales of human activity and, crucially, that it can be successfully resolved when agents give the right incentives to *alter* the objective of other agents, in such a way that the recipients' behavior changes for everyone's advantage. Indeed, a significant amount of individual, group, and international effort is expended on creating effective incentives or sanctions to shape the behavior of other individuals, social groups, and nations [39, 8, 9]. The rich body of work on game-theoretic side payments [19, 16, 14] further attests to the importance of inter-agent incentivization in society.

Translated to the framework of Markov games for multi-agent reinforcement learning (MARL) [26], the key insight is to remove the constraints of an immutable reward function. Instead, we allow agents to *learn* an incentive function that gives rewards to other learning agents and thereby shape their behavior. The new learning problem for an agent becomes two-fold: learn a policy that optimizes the total extrinsic rewards and incentives it receives, and learn an incentive function that alters other agents' behavior so as to optimize its own extrinsic objective. While the emergence of incentives in nature may have an evolutionary explanation [15], human societies contain ubiquitous examples of learned incentivization and we focus on the learning viewpoint in this work.

**The Escape Room game.** We may illustrate the benefits and necessity of incentivization with a simple example. The *Escape Room* game ER$(N, M)$ is a discrete $N$-player Markov game with individual extrinsic rewards and parameter $M < N$, as shown in Figure 1. An agent gets +10 extrinsic reward for exiting a door and ending the game, but the door can only be opened when $M$ other agents cooperate to pull the lever. However, an extrinsic penalty of $-1$ for any movement discourages all agents from taking the cooperative action. If agents optimize their own rewards with standard independent RL, no agent can attain positive reward, as we show in Section 5.

This game may be solved by equipping agents with the ability to incentivize other agents to pull the lever. However, we hypothesize—and confirm in experiments—that merely augmenting an agent's action space with a "give-reward" action and applying standard RL faces significant learning difficulties. Consider the case of ER$(2, 1)$: suppose we allow agent A1 an additional action that sends +2 reward to agent A2, and let it observe A2's chosen action prior to taking its own action. Assuming that A2 conducts sufficient exploration, an intelligent reward-giver should learn to use the give-reward action to incentivize A2 to pull the lever. However, RL optimizes the expected cumulative reward within *one* episode, but the effect of a give-reward action manifests in the recipient's behavior only after many learning updates that generally span *multiple* episodes. Hence, a reward-giver may not receive any feedback within an episode, much less an immediate feedback, on whether the give-reward action benefited its own extrinsic objective. Instead, we need an agent that explicitly accounts for the impact of incentives on the recipient's learning and, thereby, on its own future performance.

As a first step toward addressing these new challenges, we make the following conceptual, algorithmic, and experimental contributions. (1) We create an agent that learns an incentive function to reward other learning agents, by explicitly accounting for the impact of incentives on its own performance, through the learning of recipients. (2) Working with agents who conduct policy optimization, we derive the gradient of an agent's extrinsic objective with respect to the parameters of its incentive function. We propose an effective training procedure based on online cross-validation to update the incentive function and policy on the same time scale. (3) We show convergence to mutual cooperation in a matrix game, and experiment on a new deceptively simple *Escape Room* game, which poses significant difficulties for standard RL and action-based opponent-shaping agents, but on which our agent consistently attains the global optimum. (4) Finally, our agents discover near-optimal division of labor in the challenging and high-dimensional social dilemma problem of *Cleanup* [18]. Taken together, we believe this is a promising step toward a cooperative multi-agent future.

## 2    Related work

Learning to incentivize other learning agents is motivated by the problem of cooperation among independent learning agents in intertemporal social dilemmas (ISDs) [23], in which defection is preferable to individuals in the short term but mutual defection leads to low collective performance in the long term. Algorithms for fully-cooperative MARL [13, 31, 35] may not be applied as ISDs have mixed motives and cannot canonically be reduced to fully cooperative problems. Previous work showed that collective performance can be improved by independent agents with *intrinsic* rewards [10, 18, 42, 21, 17], which are either hand-crafted or slowly evolved based on other agents' performance and modulate each agent's own total reward. In contrast, a reward-giver's incentive function in our work is *learned* on the same timescale as policy learning and is given to, and maximized by, *other* agents. Empirical research shows that augmenting an agent's action space with a "give-reward" *action* can improve cooperation during certain training phases in ISDs [27].

Learning to incentivize is a form of opponent shaping, whereby an agent learns to influence the learning update of other agents for its own benefit. While LOLA [12] and SOS [25] exert influence via actions taken by its policy, whose effects manifest through the Markov game state transition, our proposed agent exerts direct influence via an incentive function, which is distinct from its policy and which explicitly affects the recipient agent's learning update. Hence the need to influence other agents does not restrict a reward-giver's policy, potentially allowing for more flexible and stable shaping. We describe the mathematical differences between our method and LOLA in Section 3.1, and experimentally compare with LOLA agents augmented with reward-giving actions.

Our work is related to a growing collection of work on modifying or learning a reward function that is in turn maximized by another learning algorithm [5, 34, 44]. Previous work investigate the evolution of the prisoner's dilemma payoff matrix when altered by a "mutant" player who gives a fixed incentive for opponent cooperation [2]; employ a centralized operator on utilities in 2-player games with side payments [34]; and directly optimize collective performance by centralized rewarding in 2-player matrix games [5]. In contrast, we work with $N$-player Markov games with self-interested agents who must individually learn to incentivize other agents and cannot optimize collective performance directly. Our technical approach is inspired by online cross validation [36], which is used to optimize hyperparameters in meta-gradient RL [43], and by the optimal reward framework [33], in which a single agent learns an intrinsic reward by ascending the gradient of its own extrinsic objective [44].

## 3    Learning to incentivize others

We design Learning to Incentivize Others (LIO), an agent that learns an incentive function by explicitly accounting for its impact on recipients' behavior, and through them, the impact on its own extrinsic objective. For clarity, we describe the ideal case where agents have a perfect model of other agents' parameters and gradients; afterwards, we remove this assumption via opponent modeling. We present the general case of $N$ LIO agents, indexed by $i \in [N] := \{1, \ldots, N\}$. Each agent gives rewards using its incentive function and learns a regular policy with all received rewards. For clarity, we use index $i$ when referring to the reward-giving part of an agent, and we use $j$ for the part that learns from received rewards. For each agent $i$, let $o^i := O^i(s) \in \mathcal{O}$ denote its individual observation at global state $s$; $a^i \in \mathcal{A}^i$ its action; and $-i$ a collection of all indices except $i$. Let $\mathbf{a}$ and $\boldsymbol{\pi}$ denote the joint action and the joint policy over all agents, respectively.

A reward-giver agent $i$ learns a vector-valued incentive function $r_{\eta^i} : \mathcal{O} \times \mathcal{A}^{-i} \mapsto \mathbb{R}^{N-1}$, parameterized by $\eta^i \in \mathbb{R}^n$, that maps its own observation $o^i$ and all other agents' actions $a^{-i}$ to a vector of rewards for the other $N-1$ agents[3]. Let $r_{\eta^i}^j$ denote the reward that agent $i$ gives to agent $j$. As we elaborate below, $r_{\eta^i}$ is separate from the agent's conventional policy and is learned via direct gradient ascent on the agent's own extrinsic objective, involving its effect on all other agents' policies, instead of via RL. Therefore, while it may appear that LIO has an augmented action space that provides an additional channel of influence on other agents, we emphasize that LIO's learning approach does *not* treat the incentive as a standard "give-reward" action.

We build on the idea of online cross-validation [36], to capture the fact that an incentive has measurable effect only after a recipient's learning step. As such, we describe LIO in a procedural manner below

(Algorithm 1). This procedure can also be viewed as an iterative method for a bilevel optimization problem [7], where the upper level optimizes the incentive function by accounting for recipients' policy optimization at the lower level. At each time step $t$, each recipient $j$ receives a total reward

$$r^j(s_t, \mathbf{a}_t, \eta^{-j}) := r^{j,\text{env}}(s_t, \mathbf{a}_t) + \sum_{i \neq j} r^j_{\eta^i}(o^i_t, a^{-i}_t), \tag{1}$$

where $r^{j,\text{env}}$ denotes agent $j$'s extrinsic reward. Each agent $j$ learns a standard policy $\pi^j$, parameterized by $\theta^j \in \mathbb{R}^m$, to maximize the objective

$$\max_{\theta^j} J^{\text{policy}}(\theta^j, \eta^{-j}) := \mathbb{E}_{\boldsymbol{\pi}}\left[\sum_{t=0}^{T} \gamma^t r^j(s_t, \mathbf{a}_t, \eta^{-j})\right]. \tag{2}$$

Upon experiencing a trajectory $\tau^j := (s_0, \mathbf{a}_0, r^j_0, \ldots, s_T)$, the recipient carries out an update

$$\hat{\theta}^j \leftarrow \theta^j + \beta f(\tau^j, \theta^j, \eta^{-j}) \tag{3}$$

that adjusts its policy parameters with learning rate $\beta$ (Algorithm 1, lines 4-5). Assuming policy optimization learners in this work and choosing policy gradient for exposition, the update function is

$$f(\tau^j, \theta^j, \eta^{-j}) = \sum_{t=0}^{T} \nabla_{\theta^j} \log \pi^j(a^j_t | o^j_t) G^j_t(\tau^j; \eta^{-j}), \tag{4}$$

where the return $G^j_t(\tau^j, \eta^{-j}) = \sum_{l=t}^{T} \gamma^{l-t} r^j(s_l, \mathbf{a}_l, \eta^{-j})$ depends on incentive parameters $\eta^{-j}$.

After each agent has updated its policy to $\hat{\pi}^j$, parameterized by new $\hat{\theta}^j$, it generates a new trajectory $\hat{\tau}^j$. Using these trajectories, each reward-giver $i$ updates its individual incentive function parameters $\eta^i$ to maximize the following individual objective (Algorithm 1, lines 6-7):

$$\max_{\eta^i} J^i(\hat{\tau}^i, \tau^i, \hat{\boldsymbol{\theta}}, \eta^i) := \mathbb{E}_{\hat{\boldsymbol{\pi}}}\left[\sum_{t=0}^{T} \gamma^t \hat{r}^{i,\text{env}}_t\right] - \alpha L(\eta^i, \tau^i). \tag{5}$$

The first term is the expected extrinsic return of the reward-giver in the new trajectory $\hat{\tau}^i$. It implements the idea that the purpose of agent $i$'s incentive function is to alter other agents' behavior so as to maximize its extrinsic rewards. The rewards it received from others are already accounted by its own policy update. The second term is a cost for giving rewards in the first trajectory $\tau^i$:

$$L(\eta^i, \tau^i) := \sum_{(o^i_t, a^{-i}_t) \in \tau^i} \gamma^t \|r_{\eta^i}(o^i_t, a^{-i}_t)\|_1. \tag{6}$$

This cost is incurred by the incentive function and not by the policy, since the latter does not determine incentivization[4] and should not be penalized for the incentive function's behavior (see Appendix A.1 for more discussion). We use the $\ell_1$-norm so that cost has the same physical "units" as extrinsic rewards. The gradient of (6) is directly available, assuming $r_{\eta^i}$ is a known function approximator (e.g., neural network). Letting $J^i(\hat{\tau}^i, \hat{\boldsymbol{\theta}})$ denote the first term in (5), the gradient w.r.t. $\eta^i$ is:

$$\nabla_{\eta^i} J^i(\hat{\tau}^i, \hat{\boldsymbol{\theta}}) = \sum_{j \neq i} (\nabla_{\eta^i} \hat{\theta}^j)^T \nabla_{\hat{\theta}^j} J^i(\hat{\tau}^i, \hat{\boldsymbol{\theta}}). \tag{7}$$

The first factor of each term in the summation follows directly from (3) and (4):

$$\nabla_{\eta^i} \hat{\theta}^j = \beta \sum_{t=0}^{T} \nabla_{\theta^j} \log \pi^j(a^j_t | o^j_t) \left(\nabla_{\eta^i} G^j_t(\tau^j; \eta^{-j})\right)^T. \tag{8}$$

Note that (3) does not contain recursive dependence of $\theta^j$ on $\eta^i$ since $\theta^j$ is a function of incentives in *previous* episodes, not those in trajectory $\tau^i$. The second factor in (7) can be derived as

$$\nabla_{\hat{\theta}^j} J^i(\hat{\tau}^i, \hat{\boldsymbol{\theta}}) = \mathbb{E}_{\hat{\boldsymbol{\pi}}}\left[\nabla_{\hat{\theta}^j} \log \hat{\pi}^j(\hat{a}^j | \hat{o}^j) Q^{i,\hat{\boldsymbol{\pi}}}(\hat{s}, \hat{\mathbf{a}})\right]. \tag{9}$$

**Algorithm 1** Learning to Incentivize Others
---
1: **procedure** TRAIN LIO AGENTS
2:     Initialize all agents' policy parameters $\theta^i$, incentive function parameters $\eta^i$
3:     **for** each iteration **do**
4:         Generate a trajectory $\{\tau^j\}$ using $\boldsymbol{\theta}$ and $\boldsymbol{\eta}$
5:         For all reward-recipients $j$, update $\hat{\theta}^j$ using (3)
6:         Generate a new trajectory $\{\hat{\tau}^i\}$ using new $\hat{\boldsymbol{\theta}}$
7:         For reward-givers $i$, compute new $\hat{\eta}^i$ by gradient ascent on (5)
8:         $\theta^i \leftarrow \hat{\theta}^i, \eta^i \leftarrow \hat{\eta}^i$ for all $i \in [N]$.
9:     **end for**
10: **end procedure**
---

In practice, to avoid manually computing the matrix-vector product in (7), one can define the loss

$$\text{Loss}(\eta^i, \hat{\tau}^i) := -\sum_{j \neq i} \sum_{t=0}^{T} \log \pi_{\hat{\theta}^j}(\hat{a}_t^j | \hat{o}_t^j) \sum_{l=t}^{T} \gamma^{l-t} r^{i,\text{env}}(\hat{s}_l, \hat{\mathbf{a}}_l), \tag{10}$$

and directly minimize it via automatic differentiation [1]. Crucially, $\hat{\theta}^j$ must preserve the functional dependence of the policy update step (4) on $\eta^i$ within the same computation graph. Derivations of (9) and (10) are similar to that for policy gradients [38] and are provided in Appendix C.

LIO is compatible with the goal of achieving emergent cooperation in fully-decentralized MARL, as agents already learn individual sets of parameters to maximize individual objectives. One may directly apply opponent modeling [3] when LIO can observe, or estimate, other agents' egocentric observations, actions, and individual rewards, and have common knowledge that all agents conduct policy updates via reinforcement learning. These requirements are satisfied in environments where incentivization itself is feasible, since these observations are required for rational incentivization. LIO may then fit a behavior model for each opponent, create an internal model of other agents' RL processes, and learn the incentive function by differentiating through fictitious updates using the model in place of (3). We demonstrate a fully-decentralized implementation in our experiments.

### 3.1 Relation to opponent shaping via actions

LIO conducts opponent shaping via the incentive function. This resembles LOLA [12], but there are key algorithmic differences. Firstly, LIO's incentive function is trained separately from its policy parameters, while opponent shaping in LOLA depends solely on the policy. Secondly, the LOLA gradient correction for agent $i$ is derived from $\nabla_{\theta^i} J^i(\theta^i, \theta^j + \Delta\theta^j)$ under Taylor expansion, but LOLA disregards a term with $\nabla_{\theta^i} \nabla_{\theta^j} J^i(\theta^i, \theta^j)$ even though it is non-zero in general. In contrast, LIO is constructed from the principle of online cross-validation [36], not Taylor expansion, and hence this particular mixed derivative is absent—the analogue for LIO would be $\nabla_{\eta^i} \nabla_{\theta^j} J^i(\theta^i, \theta^j)$, which is zero because incentive parameters $\eta^i$ affect all agents *except* agent $i$. Thirdly, LOLA optimizes its objective assuming one step of opponent learning, *before* the opponent actually does so [25]. In contrast, LIO updates the incentive function *after* recipients carry out policy updates using received incentives. This gives LIO a more accurate measurement of the impact of incentives, which reduces variance and increases performance, as we demonstrate experimentally in Appendix E.1 by comparing with a 1-episode variant of LIO that does not wait for opponent updates. Finally, by adding differentiable reward channels to the environment, which is feasible in many settings with side payments [19], LIO is closer in spirit to the paradigm of optimized rewards [33, 20, 42].

### 3.2 Analysis in Iterated Prisoner's Dilemma

LIO poses a challenge for theoretical analysis in general Markov games because each agent's policy and incentive function are updated using different trajectories but are coupled through the RL updates of all other agents. Nonetheless, a complete analysis of exact LIO—using closed-form gradient ascent without policy

Table 1: Prisoner's Dilemma

| A1/A2 | C | D |
|---|---|---|
| C | (-1, -1) | (-3, 0) |
| D | (0, -3) | (-2, -2) |

gradient approximation—is tractable in repeated matrix games. In the stateless Iterated Prisoner's Dilemma (IPD), for example, with payoff matrix in Table 1, we prove in Appendix B the following:

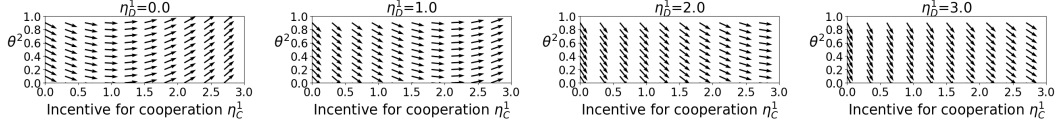

Figure 2: Exact LIO in IPD: probability of recipient cooperation versus incentive for cooperation.

**Proposition 1.** *Two LIO agents converge to mutual cooperation in the Iterated Prisoner's Dilemma.*

Moreover, we may gain further insight by visualizing the learning dynamics of exact LIO in the IPD, computed in Appendix B. Let $\eta^1 := [\eta^1_C, \eta^1_D] \in [0,3]^2$ be the incentives that Agent 1 gives to Agent 2 for cooperation (C) and defection (D), respectively. Let $\theta^2$ denote Agent 2's probability of cooperation. In Figure 2, the curvature of vector fields shows guaranteed increase in probability of recipient cooperation $\theta^2$ (vertical axis) along with increase in incentive value $\eta^1_C$ received for cooperation (horizontal axis). For higher values of incentive for defection $\eta^1_D$, greater values of $\eta^1_C$ are needed for $\theta^2$ to increase. Figure 7 shows that incentive for defection is guaranteed to decrease.

## 4 Experimental setup

Our experiments[5] demonstrate that LIO agents are able to reach near-optimal individual performance by incentivizing other agents in cooperation problems with conflicting individual and group utilities. We define three different environments with increasing complexity in Section 4.1 and describe the implementation of our method and baselines in Section 4.2.

### 4.1 Environments

**Iterated Prisoner's Dilemma (IPD).** We test LIO on the memory-1 IPD as defined in [12], where agents observe the joint action taken in the previous round and receive extrinsic rewards in Table 1. This serves as a test of our theoretical prediction in Section 3.2.

$N$**-Player Escape Room (ER).** We experiment on the $N$-player Escape Room game shown in Figure 1 (Section 1). By symmetry, any agent can receive positive extrinsic reward, as long as there are enough cooperators. Hence, for methods that allow incentivization, every agent is both a reward giver and recipient. We experiment with the cases $(N = 2, M = 1)$ and $(N = 3, M = 2)$. We also describe an asymmetric 2-player case and results in Appendix E.1.

**Cleanup.** Furthermore, we conduct experiments on the Cleanup game (Figure 3) [18, 42]. Agents get +1 individual reward by collecting apples, which spawn on the right hand side of the map at a rate that decreases linearly to zero as the amount of waste in a river approaches a depletion threshold. Each episode starts with a waste level above the threshold and no apples present. While an agent can contribute to the public good by firing a cleaning beam to clear waste, it can only do so at the river as its fixed orientation points upward. This would enable other agents to defect and selfishly collect apples, resulting in a difficult social dilemma. Each agent has an egocentric RGB image observation that spans the entire map.

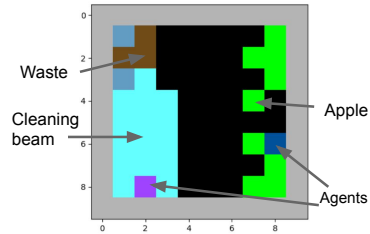

Figure 3: *Cleanup* (10x10 map): apple spawn rate decreases with increasing waste, which agents can clear with a cleaning beam.

### 4.2 Implementation and baselines

We describe key details here and provide a complete description in Appendix D.2. In each method, all agents have the same implementation without sharing parameters. The incentive function of a LIO agent is a neural network defined as follows: its input is the concatenation of the agent's observation and all other agents' chosen actions; the output layer has size $N$, sigmoid activation, and is scaled element-wise by a multiplier $R_{\max}$; each output node $j$, which is bounded in $[0, R_{\max}]$, is interpreted as the real-valued reward given to agent with index $j$ in the game (we zero-out the value it gives to itself). We chose $R_{\max} = [3, 2, 2]$ for [IPD, ER, Cleanup], respectively, so that incentives can overcome any extrinsic penalty or opportunity cost for cooperation. We use on-policy learning with policy gradient for each agent in IPD and ER, and actor-critic for Cleanup. To ensure that all agents'

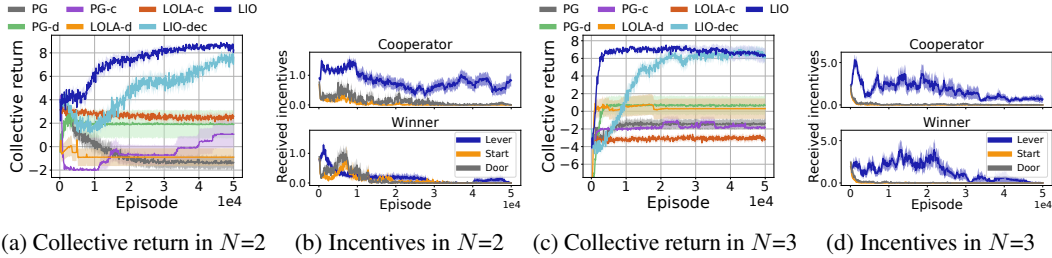

(a) Collective return in $N$=2    (b) Incentives in $N$=2    (c) Collective return in $N$=3    (d) Incentives in $N$=3

Figure 5: Escape Room. (a,c) LIO agents converge near the global optimum with value 9 (N=2) and 8 (N=3). (b,d) Incentives received for each action by the agent who ends up going to the lever/door.

policies perform sufficient exploration for the effect of incentives to be discovered, we include an exploration lower bound $\epsilon$ such that $\tilde{\pi}(a|s) = (1-\epsilon)\pi(a|s) + \epsilon/|\mathcal{A}|$, with linearly decreasing $\epsilon$.

**Fully-decentralized implementation (LIO-dec).** Each decentralized LIO agent $i$ learns a model of another agent's policy parameters $\theta^j$ via $\theta^j_{\text{estimate}} = \operatorname{argmax}_{\theta^j} \sum_{(o^j_t, a^j_t) \in \tau} \log \pi_{\theta^j}(a^j_t|o^j_t)$ at the end of each episode $\tau$. With knowledge of agent $j$'s egocentric observation and individual rewards, it conducts incentive function updates using a fictitious policy update in (3) with $\theta^j_{\text{estimate}}$ in place of $\theta^j$.

**Baselines.** The first baseline is independent policy gradient, labeled **PG**, which has the same architecture as the policy part of LIO. Second, we augment policy gradient with discrete "give-reward" actions, labeled **PG-d**, whose action space is $\mathcal{A} \times \{\text{no-op}, \text{give-reward}\}^{N-1}$. We try reward values in the set $\{2, 1.5, 1.1\}$. Giving reward incurs an equivalent cost. Next, we design a more flexible policy gradient baseline called **PG-c**, which has continuous give-reward actions. It has an augmented action space $\mathcal{A} \times [0, R_{\max}]^{N-1}$ and learns a factorized policy $\pi(a_d, a_r|o) := \pi(a_d|o)\pi(a_r|o)$, where $a_d \in \mathcal{A}$ is the regular discrete action and $a_r \in [0, R_{\max}]^{N-1}$ is a vector of incentives given to the other $N-1$ agents. Appendix D.2 describes how PG-c is trained. In ER, we run **LOLA-d** and **LOLA-c** with the same augmentation scheme as PG-d and PG-c. In Cleanup, we compare with independent actor-critic agents (**AC-d** and **AC-c**), which are analogously augmented with "give-reward" actions, and with inequity aversion (**IA**) agents [18]. We also show the approximate upper bound on performance by training a fully-centralized actor-critic (**Cen**) that is (unfairly) allowed to optimize joint reward.

## 5 Results

We find that LIO agents reach near-optimal *collective* performance in all three environments, despite being designed to optimize only *individual* rewards. This arose in ER and Cleanup because incentivization enabled agents to find an optimal division of labor[6] and in IPD where LIO is proven to converge to the CC solution. In contrast, various baselines displayed competitive behavior that led to suboptimal solutions, were not robust across random seeds, or failed to cooperate altogether. We report the results of 20 independent runs for IPD and ER, and 5 runs for Cleanup.

**Iterated Prisoner's Dilemma.** In accord with the theoretical prediction of exact LIO in Section 3.2 and Appendix B, two LIO agents with policy gradient approximation converge near the optimal CC solution with joint reward -2 in the IPD (Figure 4). This meets the performance of LOLA-PG and is close to LOLA-Ex, as reported in [12]. In the asymmetric case (LIO-asym) where one LIO agent is paired with a PG agent, we indeed find that they converge to the DC solution: PG is incentivized to cooperate while LIO defects, resulting in collective reward near -3.

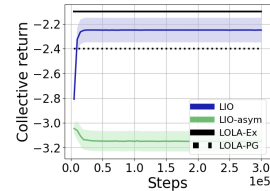

Figure 4: The sum of all agents' rewards in IPD.

**Escape Room.** Figures 5a and 5c show that groups of LIO agents discover a division of labor in both ER(2,1) and ER(3,2), whereby some agent(s) cooperate by pulling the lever to allow another agent to exit the door, such that collective return approaches the optimal value (9 for the 2-player case, 8 for the 3-player case). Fully-decentralized LIO-dec successfully solved both cases, albeit with slower learning speed. As expected, PG agents were unable to find a cooperative solution: they either stay at the start state or greedily move to the door, resulting in negative collective return. The augmented baselines PG-d and PG-c sometimes

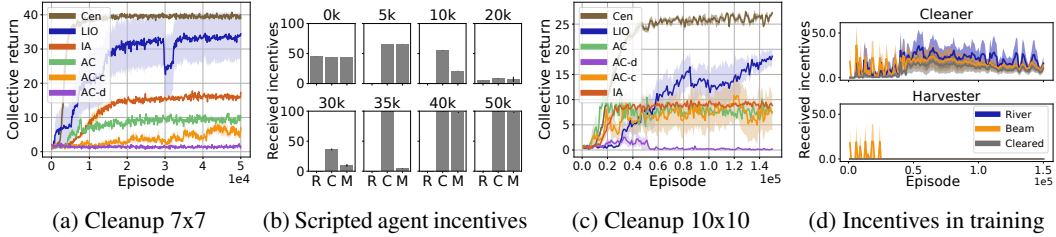

|  (a) Cleanup 7x7 | (b) Scripted agent incentives | (c) Cleanup 10x10 | (d) Incentives in training |

Figure 6: Results on Cleanup. (a,c) Emergent division of labor between LIO agents enables higher performance than AC and IA baselines, which find rewards but exhibit competitive behavior. (b) Behavior of incentive function in 7x7 Cleanup at different training checkpoints, measured against three scripted opponents: R moves within river without cleaning; C successfully cleans waste; M fires the cleaning beam but misses waste (mean and standard error of 20 evaluation episodes). (d) 10x10 map: the LIO agent who becomes a "Cleaner" receives incentives, while the "Harvester" does not.

successfully influence the learning of another agent to solve the game, but exhibit high variance across independent runs. This is strong evidence that conventional RL alone is not well suited for learning to incentivize, as the effect of "give-reward" actions manifests only in future episodes. LOLA succeeds sometimes but with high variance, as it does not benefit from the stabilizing effects of online cross-validation and separation of the incentivization channel from regular actions. Appendix E.1 contains results in an asymmetric case (LIO paired with PG), where we compare to an additional heuristic two-timescale baseline and a variant of LIO. Figure 10c evidences that LIO scales well to larger groups such as ER(5,3), since the complexity of (7) is linear in number of agents.

To understand the behavior of LIO's incentive function, we classify each agent *at the end of training* as a "Cooperator" or "Winner" based on whether its final policy has greater probability of going to the lever or door, respectively. For each agent type, aggregating over all agents of that type, we measure incentives received by that agent type when it takes each of the three actions during training. Figures 5b and 5d show that the Cooperator was correctly incentivized for pulling the lever and receives negligible incentives for noncooperative actions. Asymptotically, the Winner receives negligible incentives from the Cooperator(s), who learned to avoid the cost for incentivization (6) when doing so has no benefits itself, whereas incentives are still nonzero for the Cooperator.

**Cleanup.** Figures 6a and 6c show that LIO agents collected significantly more extrinsic rewards than AC and IA baselines in Cleanup, and approach the upper bound on performance as indicated by Cen, on both a 7x7 map and a 10x10 map with more challenging depletion threshold and lower apple respawn rates. LIO agents discovered a division of labor (Figure 11a), whereby one agent specializes to cleaning waste at the river while the other agent, who collects almost all of the apples, provides incentives to the former. In contrast, AC baselines learned clean but subsequently compete to collect apples, which is suboptimal for the group (Figure 11b). Due to continual exploration by all agents, an agent may change its behavior if it receives incentives for "wrong actions": e.g., near episode 30k in Figure 6a, an agent temporarily stopped cleaning the river despite having consistently done so earlier.

We can further understand the progression of LIO's incentive function during training as follows. First, we classify LIO agents *at the end of training* as a "Cleaner" or a "Harvester", based on whether it primarily cleans waste or collects apples, respectively. Next, we define three hand-scripted agents: an R agent moves in the river but does not clean, a C agent successfully cleans waste, and an M agent fires the cleaning beam but misses waste. Figure 6b shows the incentives given by a Harvester to these scripted agents when they are tested together periodically during training. At episodes 10k, 30k and 35k, it gave significantly more incentives to C than to M, meaning that it distinguished between successful and unsuccessful cleaning, which explains how its actual partner in training was incentivized to become a Cleaner. After 40k episodes, it gives nonzero reward for "fire cleaning beam but miss", likely because its actual training partner already converged to successful cleaning (Figure 6a), so it may have "forgotten" the difference between successful and unsuccessful usage of the cleaning beam. As shown by results in the Escape Room (Figures 5b and 5d), correct incentivization can be maintained if agents have a sufficiently large lower bound on exploration rates that pose the risk of deviating from cooperative behavior. Figure 6d shows the actual incentives received by Cleaner and Harvester agents when they are positioned in the river, fire the cleaning beam, or successfully clear waste during training. We see that asymptotically, only Harvesters provide incentives to Cleaners and not the other way around.

# 6    Conclusion and future directions

We created Learning to Incentivize Others (LIO), an agent who learns to give rewards directly to other RL agents. LIO learns an incentive function by explicitly accounting for the impact of incentives on its own extrinsic objective, through the learning updates of reward recipients. In the Iterated Prisoner's Dilemma, an illustrative *Escape Room* game, and a benchmark social dilemma problem called *Cleanup*, LIO correctly incentivizes other agents to overcome extrinsic penalties so as to discover cooperative behaviors, such as division of labor, and achieve near-optimum collective performance. We further demonstrated the feasibility of a fully-decentralized implementation of LIO.

Our approach to the goal of ensuring cooperation in a decentralized multi-agent population poses many open questions. 1) How should one analyze the simultaneous processes of learning incentive functions, which continuously modifies the set of equilibria, and learning policies with these changing rewards? While previous work have treated the convergence of gradient-based learning in differentiable games with fixed rewards [4, 25], the theoretical analysis of learning processes that dynamically change the reward structure of a game deserves more attention. 2) How can an agent account for the cost of incentives in an adaptive way? An improvement to LIO would be a handcrafted or learned mechanism that prevents the cost from driving the incentive function to zero before the effect of incentives on other agents' learning is measurable. 3) How should agents better account for the longer-term effect of incentives? One possibility is to differentiate through a sequence of gradient descent updates by recipients, during which the incentive function is fixed. 4) Should social factors modulate the effect of incentives in an agent population? LIO assumes that recipients cannot reject an incentive, but a more intelligent agent may selectively accept a subset of incentives based on its appraisal of the other agents' behavior. We hope our work sparks further interest in this research endeavor.

## Broader Impact

Our work is a step toward the goal of ensuring the common good in a potential future where independent reinforcement learning agents interact with one another and/or with humans in the real world. We have shown that cooperation can emerge by introducing an additional learned incentive function that enables one agent to affect another agent's reward directly. However, as agents still independently maximize their own individual rewards, it is open as to how to prevent an agent from misusing the incentive function to exploit others. One approach for future research to address this concern is to establish new connections between our work and the emerging literature on reward tampering [11]. By sparking a discussion on this important aspect of multi-agent interaction, we believe our work has a positive impact on the long-term research endeavor that is necessary for RL agents to be deployed safely in real-world applications.

## Acknowledgements

We thank Thomas Anthony, Jan Balaguer, and Thore Graepel at DeepMind for insightful discussions and feedback. JY was funded in part by NSF III-1717916.

## Footnotes

[3]We do not allow LIO to reward itself, as our focus is on influencing *other* agents' behavior. Nonetheless, LIO may be complemented by other methods for learning *intrinsic* rewards [44].

[4]Note that the outputs of the incentive function and policy are conditionally independent given the agent's observation, but their separate learning processes are coupled via the learning process of other agents.

[5]Code for all experiments is available at https://github.com/011235813/lio

[6]Learned behavior in Cleanup can be viewed at `https://sites.google.com/view/neurips2020-lio`

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
