[Supplementary Material]

# A Further discussion

## A.1 Cost for incentivization

We justify the way in which LIO accounts for the cost of incentivization as follows. Recall that this cost is incurred in the objective for LIO's incentive function (see (5) and (6)), instead of being accounted in the total reward (1) that is maximized by LIO's policy. Fundamentally, the reason is that the cost should be incurred only by the part of the agent that is directly responsible for incentivization. In LIO, the policy and incentive function are separate modules: while the former takes regular actions to maximize *external* rewards, only the latter produces incentives that directly and actively shape the behavior of other agents. The policy is decoupled from incentivization, and it would be incorrect to penalize it for the behavior of the incentive function. Instead, we need to attribute the cost directly to the incentive function parameters via (6). From a more intuitive perspective, LIO is constructed with the knowledge that it can perform two fundamentally different behaviors—1) take regular actions that affect the Markov game transition, and 2) give incentives to shape other agents' learning—and it knows not to penalize the former behavior with the latter behavior. In contrast, if one were to augment conventional RL with reward-giving actions (as we do for baselines in Section 4.2), then the cost for incentivization should indeed be accounted by the policy. One may consider other mechanisms for cost, such as budget constraints [27].

In our experiments, we find the coefficient $\alpha$ in the cost for incentivization is a sensitive parameter. At the beginning of training, (6) immediately drives the magnitude of incentives to zero. However, both the reward-giver and recipients require sufficient time to learn the effect of incentives, which means that too large an $\alpha$ would lead to the degenerate result of $r_{\eta^i} = \mathbf{0}$. On the other extreme, $\alpha = 0$ means there is no penalty and may result in profligate incentivization that serves no useful purpose. While we found that values of $10^{-3}$ and $10^{-4}$ worked well in our experiments, one may consider adaptive and dynamic computation of $\alpha$ for more efficient training.

# B Analysis in Iterated Prisoner's Dilemma

**Proposition 1.** *Two LIO agents converge to mutual cooperation in the Iterated Prisoner's Dilemma.*

*Proof.* We prove this by deriving closed-form expressions for the updates to parameters of policies and incentive functions. These updates are also used to compute the vector fields shown in Figure 2. Let $\theta^i$ for $i \in \{1, 2\}$ denote each agent's probability of taking the cooperative action. Let $\eta^1 := [\eta_C^1, \eta_D^1] \in \mathbb{R}^2$ denote Agent 1's incentive function, where the values are given to Agent 2 when it takes action $a^2 = C$ or $a^2 = D$. Similarly, let $\eta^2$ denote Agent 2's incentive function. The value function for each agent is defined by

$$V^i(\theta^1, \theta^2) = \sum_{t=0}^{\infty} \gamma^t p^T r^i = \frac{1}{1-\gamma} p^T r^i,$$

$$\text{where} \quad p = \left[\theta^1\theta^2, \theta^1(1-\theta^2), (1-\theta^1)\theta^2, (1-\theta^1)(1-\theta^2)\right].$$

$$(11)$$

The total reward received by each agent is

$$r^1 = \left[-1 + \eta_C^2, -3 + \eta_C^2, 0 + \eta_D^2, -2 + \eta_D^2\right], \tag{12}$$

$$r^2 = \left[-1 + \eta_C^1, 0 + \eta_D^1, -3 + \eta_C^1, -2 + \eta_D^1\right]. \tag{13}$$

Agent 2 updates its policy via the update

$$\begin{aligned}
\hat{\theta}^2 &= \theta^2 + \alpha \nabla_{\theta^2} V^2(\theta^1, \theta^2) \\
&= \theta^2 + \frac{\alpha}{1-\gamma} \nabla_{\theta^2} \left(\theta^1\theta^2(-1 + \eta_C^1) + \theta^1(1-\theta^2)\eta_D^1 \right. \\
&\quad \left. + (1-\theta^1)\theta^2(-3 + \eta_C^1) + (1-\theta^1)(1-\theta^2)(-2 + \eta_D^1)\right) \\
&= \theta^2 + \frac{\alpha}{1-\gamma}\left(\eta_C^1 - \eta_D^1 - 1\right),
\end{aligned} \tag{14}$$

and likewise for Agent 1:

$$\hat{\theta}^1 = \theta^1 + \frac{\alpha}{1-\gamma}\left(\eta_C^2 - \eta_D^2 - 1\right). \tag{15}$$

Figure 7: Vector fields showing the probability of recipient cooperation versus incentive value given for cooperation (top row) and defection (lower row). Each plot has a fixed value for the incentive given for the other action.

Let $\hat{p}$ denote the joint action probability under updated policies $\hat{\theta}^1$ and $\hat{\theta}^2$, and let $\Delta^2 := (\eta_C^1 - \eta_D^1 - 1)\alpha/(1-\gamma)$ denote Agent 2's policy update. Agent 1 updates its incentive function parameters via

$$
\begin{aligned}
\eta^1 &\leftarrow \eta^1 + \beta \nabla_{\eta^1} \frac{1}{1-\gamma} \hat{p}^T r^1 \\
&= \eta^1 + \frac{\beta}{1-\gamma} \nabla_{\eta^1} \Big[ \hat{\theta}^1(\theta^2 + \Delta^2)(-1 + \eta_C^2) + \hat{\theta}^1(1 - \theta^2 - \Delta^2)(-3 + \eta_C^2) \\
&\quad + (1 - \hat{\theta}^1)(\theta^2 + \Delta^2)\eta_D^2 + (1 - \hat{\theta}^1)(1 - \theta^2 - \Delta^2)(-2 + \eta_D^2) \Big] \\
&= \eta^1 + \frac{\beta\alpha}{(1-\gamma)^2} B_2 \begin{bmatrix} 1 \\ -1 \end{bmatrix},
\end{aligned}
\tag{16}
$$

where the scalar $B_2$ is

$$
B_2 = \hat{\theta}^1(-1 + \eta_C^2) - \hat{\theta}^1(-3 + \eta_C^2) + (1 - \hat{\theta}^1)\eta_D^2 - (1 - \hat{\theta}^1)(-2 + \eta_D^2) = 2.
\tag{17}
$$

By symmetry, with $B_1 = 2$, Agent 2 updates its incentive function via

$$
\eta^2 \leftarrow \eta^2 + \frac{\beta\alpha}{(1-\gamma)^2} B_1 \begin{bmatrix} 1 \\ -1 \end{bmatrix}.
\tag{18}
$$

Note that each $\eta^i$ is updated so that $\eta_C^i$ increases while $\eta_D^i$ decreases. Referring to (14) and (15), one sees that the updates to incentive parameters lead to updates to policy parameters that increase the probability of mutual cooperation. This is consistent with the viewpoint of modifying the Nash Equilibrium of the payoff matrices. With incentives, the players have payoff matrices in Table 2. For CC to be the global Nash Equilibrium, such that cooperation is preferred by an agent $i$ regardless of the other agent's action, incentives must satisfy $\eta_C^i - \eta_D^i - 1 > 0$. This is guaranteed to occur by incentive updates (16) and (18). $\qquad\square$

Table 2: Payoff matrices for row player (left) and column player (right) with incentives.

| A1 | C | D |
|---|---|---|
| C | $-1 + \eta_C^2$ | $-3 + \eta_C^2$ |
| D | $0 + \eta_D^2$ | $-2 + \eta_D^2$ |

| A2 | C | D |
|---|---|---|
| C | $-1 + \eta_C^1$ | $0 + \eta_D^1$ |
| D | $-3 + \eta_C^1$ | $-2 + \eta_D^1$ |

## C  Derivations

The factor $\nabla_{\hat{\theta}^j} J^i(\hat{\tau}^i, \hat{\boldsymbol{\theta}})$ (9) in the incentive function's gradient (7) is derived as follows. For brevity, we will drop the "hat" notation—recall that it indicates a quantity belongs to a new trajectory after a regular policy update—as all quantities here have "hats". Let $\nabla_j$ denote $\nabla_{\hat{\theta}^j}$ and $\pi$ denote $\pi(a_t|s_t)$. Let $V^{i,\boldsymbol{\pi}}(s)$ and $Q^{i,\boldsymbol{\pi}}(s, \mathbf{a})$ denote the global value and action-value function for agent $i$'s reward

under joint policy $\boldsymbol{\pi}$. Then the gradient of agent $i$'s expected extrinsic return with respect to agent $j$'s policy parameter can be derived in a similar manner as standard policy gradients [38]:

$$\nabla_j J^i(\tau, \boldsymbol{\theta}) = \nabla_j V^{i,\boldsymbol{\pi}}(s_0) = \nabla_j \sum_{\mathbf{a}} \boldsymbol{\pi}(\mathbf{a}|s_0) Q^{i,\boldsymbol{\pi}}(s_0, \mathbf{a})$$

$$= \sum_{\mathbf{a}} \pi^{-j} \left( (\nabla_j \pi^j) Q^{i,\boldsymbol{\pi}}(s_0, \mathbf{a}) + \pi^j \nabla_j Q^{i,\boldsymbol{\pi}}(s_0, \mathbf{a}) \right)$$

$$= \sum_{\mathbf{a}} \pi^{-j} \left( (\nabla_j \pi^j) Q^{i,\boldsymbol{\pi}} + \pi^j \nabla_j \left( r^i + \gamma \sum_{s'} P(s'|s_0, \mathbf{a}) V^{i,\boldsymbol{\pi}}(s') \right) \right)$$

$$= \sum_{\mathbf{a}} \pi^{-j} \left( (\nabla_j \pi^j) Q^{i,\boldsymbol{\pi}} + \gamma \pi^j \sum_{s'} P(s'|s_0, \mathbf{a}) \nabla_j V^{i,\boldsymbol{\pi}}(s') \right)$$

$$= \sum_{x} \sum_{k=0}^{\infty} P(s_0 \to x, k, \boldsymbol{\pi}) \gamma^k \sum_{\mathbf{a}} \pi^{-j} \nabla_j \pi^j Q^{i,\boldsymbol{\pi}}(x, \mathbf{a})$$

$$= \sum_{s} d^{\boldsymbol{\pi}}(s) \sum_{\mathbf{a}} \pi^{-j} \nabla_j \pi^j Q^{i,\boldsymbol{\pi}}(s, \mathbf{a})$$

$$= \sum_{s} d^{\boldsymbol{\pi}}(s) \sum_{\mathbf{a}} \pi^{-j} \pi^j \nabla_j \log \pi^j Q^{i,\boldsymbol{\pi}}(s, \mathbf{a})$$

$$= \mathbb{E}_{\boldsymbol{\pi}} \left[ \nabla_j \log \pi^j(a^j|s) Q^{i,\boldsymbol{\pi}}(s, \mathbf{a}) \right]$$

Alternatively, one may rely on automatic differentiation in modern machine learning frameworks [1] to compute the chain rule (7) via direct minimization of the loss (10). This is derived as follows. Let the notation $\neq j, i$ denote all indices except $j$ and $i$. Note that agent $i$'s updated policy $\hat{\pi}^i$ is not a function of $\eta^i$, as it does not receive incentives from itself. Recall that a recipient $j$'s updated policy $\hat{\pi}^j$ has explicit dependence on a reward-giver $i$'s incentive parameters $\eta^i$. Also note that

$$\nabla_{\eta^i} \hat{\pi}^{-i} = \sum_{j \neq i} (\nabla_{\eta^i} \hat{\pi}^j) \hat{\pi}^{\neq j, i}$$

by the product rule. Then we have:

$$\nabla_{\eta^i} J^i(\hat{\tau}^i, \hat{\boldsymbol{\theta}}) = \nabla_{\eta^i} V^{i,\hat{\boldsymbol{\pi}}}(\hat{s}_0) = \nabla_{\eta^i} \sum_{\hat{\mathbf{a}}} \hat{\pi}^i(\hat{a}^i|\hat{s}_0) \hat{\pi}^{-i}(\hat{a}^{-i}|\hat{s}_0) Q^{i,\hat{\boldsymbol{\pi}}}(\hat{s}_0, \hat{\mathbf{a}})$$

$$= \sum_{\hat{\mathbf{a}}} \hat{\pi}^i \left( \sum_{j \neq i} (\nabla_{\eta^i} \hat{\pi}^j) \hat{\pi}^{\neq j, i} Q^{i,\hat{\boldsymbol{\pi}}} + \hat{\pi}^{-i} \nabla_{\eta^i} Q^{i,\hat{\boldsymbol{\pi}}} \right) \quad \text{(by the remarks above)}$$

$$= \sum_{\hat{\mathbf{a}}} \hat{\pi}^i \left( \sum_{j \neq i} (\nabla_{\eta^i} \hat{\pi}^j) \hat{\pi}^{\neq j, i} Q^{i,\hat{\boldsymbol{\pi}}} + \gamma \hat{\pi}^{-i} \sum_{s'} P(s'|\hat{s}_0, \hat{\mathbf{a}}) \nabla_{\eta^i} V^{i,\hat{\boldsymbol{\pi}}}(s') \right)$$

$$= \sum_{x} \sum_{k=0}^{\infty} P(s_0 \to x, k, \hat{\boldsymbol{\pi}}) \gamma^k \sum_{\hat{\mathbf{a}}} \hat{\pi}^i \sum_{j \neq i} (\nabla_{\eta^i} \hat{\pi}^j) \hat{\pi}^{\neq j, i} Q^{i,\hat{\boldsymbol{\pi}}}$$

$$= \sum_{\hat{s}} d^{\hat{\boldsymbol{\pi}}}(\hat{s}) \sum_{\hat{\mathbf{a}}} \hat{\pi}^i \sum_{j \neq i} \hat{\pi}^j (\nabla_{\eta^i} \log \hat{\pi}^j) \hat{\pi}^{\neq j, i} Q^{i,\hat{\boldsymbol{\pi}}}$$

$$= \sum_{\hat{s}} d^{\hat{\boldsymbol{\pi}}}(\hat{s}) \sum_{\hat{\mathbf{a}}} \hat{\pi}^i \sum_{j \neq i} (\nabla_{\eta^i} \log \hat{\pi}^j) \hat{\pi}^{-i} Q^{i,\hat{\boldsymbol{\pi}}}$$

$$= \sum_{\hat{s}} d^{\hat{\boldsymbol{\pi}}}(\hat{s}) \sum_{\hat{\mathbf{a}}} \hat{\boldsymbol{\pi}} \sum_{j \neq i} (\nabla_{\eta^i} \log \hat{\pi}^j) Q^{i,\hat{\boldsymbol{\pi}}} = \mathbb{E}_{\hat{\boldsymbol{\pi}}} \left[ \sum_{j \neq i} (\nabla_{\eta^i} \log \hat{\pi}^j) Q^{i,\hat{\boldsymbol{\pi}}} \right]$$

Hence descending a stochastic estimate of this gradient is equivalent to minimizing the loss in (10).

# D  Experiments

## D.1  Environment details

This section provides more details on each experimental setup.

**IPD.** We used the same definition of observation, action, and rewards as Foerster *et al.* [12]. Each environment step is one round of the matrix game. Each agent observes the joint action taken by both agents at the previous step, along with an indicator for the first round of each episode. We trained for 60k episodes, each with 5 environments steps, which gives the same total number of environment steps used by LOLA [12].

**Escape Room.** Each agent observes all agents' positions and can move among the three available states: lever, start, and door. At every time step, all agents commit to and disclose their chosen actions, compute the incentives based on their observations of state and others' actions (only for LIO and augmented baselines that allow incentivization), and receive the sum of extrinsic rewards and incentives (if any). LIO and augmented baselines also observe the cumulative incentives given to the other agents within the current episode. An agent's individual reward is zero for staying at the current state, -1 for movement away from its current state if fewer than $M$ agents move to (or are currently at) the lever, and +10 for moving to (or staying at) the door if $\geq M$ agents pull the lever. Each episode terminates when an agent successfully exits the door, or when 5 time steps elapse.

**Cleanup.** We built on a version of an open-source implementation [40]. The environment settings for 7x7 and 10x10 maps are given in Table 3. To focus on the core aspects of the common-pool resource problem, we removed rotation actions, set the orientation of all agents to face "up", and disabled their "tagging beam" (which, if used, would remove a tagged agent from the environment for a number of steps). These changes mean that an agent must move to the river side of the map to clear waste successfully, as it cannot simply stay in the apple patch and fire its cleaning beam toward the river. Acting cooperatively as such would allow other agents to collect apples, and hence our setup increases the difficulty of the social dilemma. Each agent receives an egocentric normalized RGB image observation that spans a sufficiently large area such that the entire map is observable by that agent regardless of its position. The cleaning beam has length 5 and width 3. For LIO and the AC-c baseline, which have a separate module that observes other agents' actions and outputs real-valued incentives, we let that module observe a multi-hot vector that indicates which agent(s) used their cleaning beam.

Table 3: Environment settings in Cleanup

| Parameter | 7x7 | 10x10 |
|---|---|---|
| appleRespawnProbability | 0.5 | 0.3 |
| thresholdDepletion | 0.6 | 0.4 |
| thresholdRestoration | 0.0 | 0.0 |
| wasteSpawnProbability | 0.5 | 0.5 |
| view_size | 4 | 7 |
| max_steps | 50 | 50 |

## D.2  Implementation

This subsection provides more details on implementation of all algorithms used in experiments. We use fully-connected neural networks for function approximation in the IPD and ER, and convolutional networks to process image observations in Cleanup. The policy network has a softmax output for discrete actions in all environments. Within each environment, all algorithms use the same neural architecture unless stated otherwise. We applied the open-source implementation of LOLA [12] to ER. We use an exploration lower bound $\epsilon$ that maps the learned policy $\pi$ to a behavioral policy $\tilde{\pi}(a|s) = (1 - \epsilon)\pi(a|s) + \epsilon/|\mathcal{A}|$, with $\epsilon$ decaying linearly from $\epsilon_{\text{start}}$ to $\epsilon_{\text{end}}$ by $\epsilon_{\text{div}}$ episodes. We use discount factor $\gamma = 0.99$. We use gradient descent for policy optimization, the Adam optimizer [22] for training value functions (in Cleanup), and Adam optimizer for LIO's incentive function.

The augmented policy gradient and actor-critic baselines, labeled as PG-c and AC-c, which have continuous "give-reward" actions in addition to regular discrete actions, are trained as follows. These

baselines have an augmented action space $\mathcal{A} \times \mathbb{R}^{N-1}$ and learns a factorized policy $\pi(a_d, a_r|o) := \pi(a_d|o)\pi(a_r|o)$, where $a_d \in \mathcal{A}$ is a regular discrete action and $a_r \in \mathbb{R}^{N-1}$ is the reward given to the other $N-1$ agents. The factor $\pi(a_d|o)$ is a standard categorical distribution conditioned on observation. The factor $\pi(a_r|o)$ is defined via an element-wise sigmoid $\sigma(\cdot)$ applied to samples from a multivariate diagonal Gaussian, so that $\pi(a_r|o)$ is bounded. Specifically, we let $u \sim \mathcal{N}(f_\eta(o), \mathbf{1})$, where $f_\eta(o) \colon \mathcal{O} \mapsto \mathbb{R}^{N-1}$ is a neural network with parameters $\eta$, and let $a_r = R_{\max}\sigma(u)$. By the change of variables formula, $\pi(a_r|o)$ has density $\pi(a_r|o) = \mathcal{N}(\mu_\eta, \mathbf{1}) \prod_{i=1}^{N-1}(\mathrm{d}a_r[i]/\mathrm{d}u[i])^{-1}$, which can be used to compute the log-likelihood of $\pi(a_d, a_r|o)$ in the policy gradient.

Let $\beta$ denote the coefficient for entropy of the policy, $\alpha_\theta$ the policy learning rate, $\alpha_\eta$ the incentive learning rate, $\alpha_\phi$ the critic learning rate, and $R_a$ the value of the discrete "give-reward" action.

**IPD.** The policy network and the incentive function in LIO have two hidden layers of size 16 and 8.

Table 4: Hyperparameters in IPD.

| Parameter | Value | Parameter | Value |
|---|---|---|---|
| $\beta$ | 0.1 | $\alpha_\theta$ | 1e-3 |
| $\epsilon_{\text{start}}$ | 1.0 | $\alpha_\eta$ | 1e-3 |
| $\epsilon_{\text{end}}$ | 0.01 | $\alpha$ | 0 |
| $\epsilon_{\text{div}}$ | 5000 | $R_{\max}$ | 3.0 |

**ER.** The policy network has two hidden layers of size 64 and 32. LIO's incentive function has two hidden layers of size 64 and 16. We use a separate Adam optimizer for the cost part of the incentive function's objective (5), with learning rate 1e-4, with $\alpha_\eta = $ 1e-3, and set $\alpha = 1.0$. Exploration and learning rate hyperparameters were tuned for each algorithm via coordinate ascent, searching through $\epsilon_{\text{start}}$ in [0.5, 1.0], $\epsilon_{\text{end}}$ in [0.05, 0.1, 0.3], $\epsilon_{\text{div}}$ in [100, 1000], $\beta$ in [0.01, 0.1], $\alpha_\theta$, $\alpha_\eta$, and $\alpha_{\text{cost}}$ in [1e-3, 1e-4]. LOLA performed best with learning rate 0.1 and $R_a = 2.0$, but it did not benefit from additional exploration. LIO and PG-c have $R_{\max} = 2.0$. PG-d used $R_a = 2.0$.

Table 5: Hyperparameters in Escape Room.

| Parameter | $N = 2$ | | | | $N = 3$ | | | |
|---|---|---|---|---|---|---|---|---|
| | LIO | PG | PG-d | PG-c | LIO | PG | PG-d | PG-c |
| $\beta$ | 0.01 | 0.01 | 0.01 | 0.1 | 0.01 | 0.01 | 0.01 | 0.1 |
| $\epsilon_{\text{start}}$ | 0.5 | 0.5 | 0.5 | 1.0 | 0.5 | 0.5 | 0.5 | 1.0 |
| $\epsilon_{\text{end}}$ | 0.1 | 0.05 | 0.05 | 0.1 | 0.3 | 0.05 | 0.05 | 0.1 |
| $\epsilon_{\text{div}}$ | 1e3 | 1e2 | 1e2 | 1e3 | 1e3 | 1e2 | 1e2 | 1e3 |
| $\alpha_\theta$ | 1e-4 | 1e-4 | 1e-4 | 1e-3 | 1e-4 | 1e-4 | 1e-4 | 1e-3 |

**Cleanup.** All algorithms are based on actor-critic for policy optimization, whereby each agent $j$'s policy parameter $\theta^j$ is updated via

$$\hat{\theta}^j \leftarrow \theta^j + \mathbb{E}_{\boldsymbol{\pi}}\left[\nabla_{\theta^j} \log \pi_{\theta^j}(a^j|o^j)\left(r^j + \gamma V_{\phi^j}(s') - V_{\tilde{\phi}^j}(s)\right)\right], \qquad (19)$$

and the critic parameter $\phi^j$ is updated by minimizing the temporal difference loss

$$L(\phi^j) = \mathbb{E}_{s,s'\sim\boldsymbol{\pi}}\left[\left(r^j + \gamma V_{\tilde{\phi}^j}(s') - V_{\phi^j}(s)\right)^2\right] \qquad (20)$$

The target network [28] parameters $\tilde{\phi}^j$ are updated via $\tilde{\phi}^j \leftarrow \tau\phi^j + (1-\tau)\tilde{\phi}^j$ with $\tau = 0.01$.

The policy and value networks have an input convolutional layer with 6 filters of size [3, 3], stride [1, 1], and ReLU activation. The output of convolution is flattened and passed through two fully-connected (FC) hidden layers both of size 64. The policy output is a softmax over discrete actions; the value network has a linear scalar output. LIO's incentive function uses the same input convolutional layer, except that its output is passed through the first FC layer, concatenated with its observation of other agents' actions, then passed through the second FC layer and finally to a linear output layer. Inequity Aversion agents [18] have an additional 1D vector observation of all agents' temporally

(a) Agent A2 incurs an extrinsic penalty for any change of state.

(b) Agent A1 is penalized at every step if A2 does not pull the lever.

(c) A1 get +10 at the door if A2 pulls the lever.

Figure 8: Asymmetric *Escape Room* game involving two agents, A1 and A2. (a) In the absence of incentives, A2's optimal policy is to stay at the start state and not pull the lever. (b) Hence A1 cannot exit the door and is penalized at every step. (c) A1 can receive positive reward if it learns to incentivize A2 to pull the lever. Giving incentives is not an action depicted here.

smoothed rewards—this is concatenated with the output of the first FC hidden layer and sent to the second FC layer. Entropy coefficient was held at 0.1 for all methods.

LIO and AC-c have $R_{\max} = 2.0$. AC-d used $R_a = 2.0$. Inequity aversion agents have disadvantageous aversion coefficient value 0, advantageous aversion coefficient value 0.05, and temporal smoothing parameter $\lambda = 0.95$. We use critic learning rate $\alpha_\phi = 10^{-3}$ for all methods. LIO used $\alpha_\eta =$1e-3 and cost coefficient $\alpha = 10^{-4}$. Exploration and learning rate hyperparameters were tuned for each algorithm via coordinate ascent, searching through $\epsilon_{\text{start}}$ in [0.5, 1.0], $\epsilon_{\text{end}}$ in [0.05, 0.1], $\epsilon_{\text{div}}$ in [100, 1000, 5000], $\alpha_\theta$, $\alpha_\eta$, and $\alpha_{\text{cost}}$ in [1e-3, 1e-4].

Table 6: Hyperparameters in Cleanup.

| Parameter | 7x7 | | | | | 10x10 | | | | |
|---|---|---|---|---|---|---|---|---|---|---|
| | LIO | AC | AC-d | AC-c | IA | LIO | AC | AC-d | AC-c | IA |
| $\epsilon_{\text{start}}$ | 0.5 | 0.5 | 0.5 | 0.5 | 0.5 | 0.5 | 0.5 | 0.5 | 0.5 | 0.5 |
| $\epsilon_{\text{end}}$ | 0.05 | 0.05 | 0.05 | 0.05 | 0.05 | 0.05 | 0.05 | 0.05 | 0.05 | 0.05 |
| $\epsilon_{\text{div}}$ | 100 | 100 | 100 | 100 | 1000 | 1000 | 5000 | 1000 | 1000 | 5000 |
| $\alpha_\theta$ | 1e-4 | 1e-3 | 1e-4 | 1e-4 | 1e-3 | 1e-4 | 1e-3 | 1e-3 | 1e-3 | 1e-3 |

# E    Additional results

## E.1    Asymmetric Escape Room

We conducted additional experiments on an asymmetric version of the Escape Room game between two learning agents (A1 and A2) as shown in Figure 8. A1 gets +10 extrinsic reward for exiting a door and ending the game (Figure 8c), but the door can only be opened when A2 pulls a lever; otherwise, A1 is penalized at every time step (Figure 8b). The extrinsic penalty for A2 discourages it from taking the cooperative action (Figure 8a). The global optimum combined reward is +9, and it is impossible for A2 to get positive extrinsic reward. Due to the asymmetry, A1 is the reward-giver and A2 is the reward recipient for methods that allow incentivization. Each agent observes both agents' positions, and can move between the two states available to itself. We allow A1 to observe A2's current action before choosing its own action, which is necessary for methods that learn to reward A2's cooperative actions. We use a standard policy gradient for A2 unless otherwise specified.

In addition to the baselines described for the symmetric case—namely, policy gradient (PG-rewards) and LOLA with discrete "give-reward" actions—we also compare with a two-timescale method, labeled **2-TS**. A 2-TS agent has the same augmented action space as the PG-rewards baseline, except that it learns over a longer time horizon than the reward recipient. Each "epoch" for the 2-TS agent spans multiple regular episodes of the recipient, during which the 2-TS agent executes a fixed policy. The 2-TS agent only caries out a learning update using a final terminal reward, which is the average extrinsic rewards it gets during *test* episodes that are conducted at the end of the epoch. Performance on test episodes serve as a measure of whether correct reward-giving actions were taken to influence the recipient's learning during the epoch. To our knowledge, 2-TS is a novel baseline but has key limitations: the use of two timescales only applies to the asymmetric 2-player game, and requires fast learning by the reward-recipient, chosen to be a tabular Q-learning, to avoid intractably long epochs.

(a) Sum of agent rewards     (b) Two PG agents     (c) LIO (A1) and PG agent (A2)     (d) 1-episode LIO and PG agent

Figure 9: Results in asymmetric 2-player Escape Room. (a) LIO (paired with PG) converges rapidly to the global optimum, 2-TS (paired with tabular Q-learner) converges slower, while policy gradient baselines could not cooperate. (b) Two PG agents cannot cooperate, as A2 converges to "do-nothing". (c) A LIO agent (A1) attains near-optimum reward by incentivizing a PG agent (A2). (d) 1-episode LIO has larger variance and lower performance. Normalization factors are 1/10 (A1) and 1/2 (A2).

Figure 9 shows the sum of both agents' rewards for all methods on the asymmetric 2-player game, as well as agent-specific performance for policy gradient and LIO, across training episodes. A LIO reward-giver agent paired with a policy gradient recipient converges rapidly to a combined return near 9.0 (Figure 9a), which is the global maximum, while both PG and PG-rewards could not escape the global minimum for A1. LOLA paired with a PG recipient found the cooperative solution in two out of 20 runs; this suggests the difficulty of using a fixed incentive value to conduct opponent shaping via discrete actions. The 2-TS method is able to improve combined return but does so much more gradually than LIO, because an epoch consists of many base episodes and it depends on a highly delayed terminal reward. Figure 9b for two PG agents shows that A2 converges to the policy of not moving (reward of 0), which results in A1 incurring penalties at every time step. In contrast, Figure 9c verifies that A1 (LIO) receives the large extrinsic reward (scaled by 1/10) for exiting the door, while A2 (PG) has average normalized reward above -0.5 (scaled by 1/2), indicating that it is receiving incentives from A1. Average reward of A2 (PG) is below 0 because incentives given by A1 need not exceed 1 continually during training—once A2's policy is biased toward the cooperative action in early episodes, its decaying exploration rate means that it may not revert to staying put even when incentives do not overcome the penalty for moving. Figure 9d shows results on a one-episode version of LIO where the same episode is used for both policy update and incentive function updates, with importance sampling corrections. This version performs significantly lower for A1 and gives more incentives than is necessary to encourage A2 to move. It demonstrates the benefit of learning the reward function using a separate episode from that in which it is applied.

## E.2 Symmetric Escape Room

Figure 10 shows total reward (extrinsic + received - given incentives), counts of "lever" and "door" actions, and received incentives in one training run each for ER(2,1) and ER(3,2). In Figure 10a, A1 becomes the winner and A2 the cooperator. It is not always necessary for A1 to give rewards. The fact that LIO models the learning updates of recipients may allow it to find that reward-giving is unnecessary during some episodes when the recipient's policy is sufficiently biased toward cooperation. In Figure 10b, A3 converges to going to the door, as it incentives A1 and A2 to pull the lever.

(a) ER(2,1)      (b) ER(3,2)      (c) ER(5,3)

Figure 10: (a,b) Individual actions and incentives in ER(2,1) and ER(3,2). (c) LIO converges to the global optimum in ER(5,3).

(a) Division of labor    (b) AC agents compete    (c) Cleaner's incentives    (d) 10x10 map

Figure 11: (a) In 7x7 Cleanup, one LIO agent learns to focus on cleaning waste, as it receives incentives from the other who only collects apple. (b) In contrast, AC agents compete for apples after cleaning. (c) Incentives received during training on 7x7 Cleanup. (d) Behavior of incentive function against scripted opponent policies on 10x10 map.

### E.3   Cleanup

Figure 11a is a snapshot of the division of labor found by two LIO agents, whereby the blue agent picks apples while the purple agent stays on the river side to clean waste. The latter does so because of incentives from the former. In contrast, Figure 11b shows a time step where two AC agents compete for apples, which is jointly suboptimal. Figure 11c shows the received incentives during training in the 7x7 map, for each of two LIO agents that were classified after training as a "Cleaner" or "Harvester". Figure 11d shows the incentives given by a "Harvester" agent to three scripted agents during each training checkpoint.

Agents with hand-designed intrinsic rewards based on social influence [21] also outperform standard RL agents on Cleanup. We can make an indirect comparison to [21] by noting that IA reaches a score around 250 by $1.6 \times 10^8$ steps [18, Figure 3a], which outperforms the score of 200 attained by Social Influence at $3 \times 10^8$ steps [21, Figure 1a] in the original Cleanup map with 5 agents. Hence, the fact that LIO outperforms IA in our experiments suggests that LIO compares favorably with Social Influence, provided that LIO uses the same RL algorithm as the latter for policy optimization.