[Reviews · NeurIPS 2020]

Review 1

Summary and Contributions: This paper presents a new framework for multi-agent reinforcement learning by allowing the agents to incentivize other agents by giving out their own "rewards". An effective algorithm is also proposed for effective policy learning within this new framework and empirical results are shown on several MARL testbeds.

Strengths: The framework of allowing agents to give out reward is a novel and interesting contribution to the whole MARL community, which has great potential for solving a wide range of problems, such as credit assignment, cooperation, emergent behavior, etc. The analysis and derivation of the algorithm are neat, clear, and insightful. In general, I like this paper.

Weaknesses: I have two concerns on (1) baselines and (2) scalability. (1) regarding the baselines, although I do think the current experiments are sound, I would be interested to see comparisons with other MARL approaches beyond the LOLA-style second-order optimization approaches. IA is a good one and it is nice to see that LIO outperforms IA, but I do think the results can be more convincing if more benchmark algorithms can be included. For example, does social influence solves the problem (https://arxiv.org/pdf/1810.08647.pdf)? Mutual information can be also viewed as an approximation of accounting other agents' future policy change and has shown great performances in harvest/cleanup with a large number of agents. Can we simply learn a value function conditioned on the received rewards of different agents (in the same spirit of DDPG) so that we can avoid performing second-order gradient? These are the questions raised when I read the paper and I believe a more in-depth discussion/experiments will further consolidate the contribution of this work. (2) regarding scalability, it is a bit unfortunate that the biggest experiment in this paper includes only 3 agents. I could totally imagine that the “bi-level” optimization scheme makes the algorithm inefficient when the total number of agents increase but it still remains critical for the readers to understand the asymptotic performance of the algorithm --- even if it doesn’t work, a discussion on the limitation and potential improvement can be extremely useful. I would strongly recommend the authors to include some analysis on environments with N>3 (e.g., N=5, 7, 10).

Correctness: The claims and methods are sound and rigorous.

Clarity: The paper is well-written and easy to follow.

Relation to Prior Work: The discussions are clear enough.

Reproducibility: Yes

Additional Feedback: I like this paper, but, as I mentioned above, I would strongly suggest the authors include more benchmarks (e.g., social influence) and results with more agents (e.g., 5 or 7 agents in cleanup like what the social influence paper did) to make the paper stronger. I can further increase my score if my concerns can be addressed. =============== After Rebuttal =============== I have checked the comments and I have increased my score based on the additional experiments with the Social Influence paper and with more agents. I would strongly encourage the author to include these two additional experiments and leave some discussions on scalability in the final version to make the paper much stronger.


Review 2

Summary and Contributions: The authors extent multi-agent social dilemma's to include a channel for sharing reward. The develop a new agent that in addition to learning to optimize it own reward, learns an influence function to shape the behavior of others. The author describe compare this approach to other agent shaping techniques (LOLA in particular) and show empirically that when these new agents are allowed to influence each other by transferring reward they reliably achieve a high collective reward.

Strengths: The paper is clearly written. The new toy environment (Escape Room) a great pedagogical and testing tool for demonstrating the power of this approach. The experiments are well executed. The quantitative and analytic results are clear and the qualitative analyses of the influence functions give good insights into the learned policies. I thought the detailed comparison to LOLA was well done and helps situate these results among that line of work. While the idea that side-payments can enable more robust cooperation in repeated games is well known the authors nicely demonstrate that naive implementations fail to realize the full potential of these methods. The introduction motivates these challenges well.

Weaknesses: I'm not sure how broadly interested the NeurIPS community will be in these results. I would like to see a greater attempt to explain specific ways that these techniques could be used in a more scaled up context. What might the currency of reward be in the real world (or even a simulated game world)? What assumptions made in LIO will not hold or require relaxing? Why did the agent start rewarding cleaning up but missing at 40K+ time steps? It is at least worth speculating in the paper.

Correctness: Yes

Clarity: Yes

Relation to Prior Work: Yes

Reproducibility: Yes

Additional Feedback: The rebuttal improves an already strong submission. Thank you for the response.


Review 3

Summary and Contributions: The paper proposes a framework where agents can shape other agents’ behaviors by directly rewarding other agents. The authors separate a task policy with the reward-giving policy. Each agent learns its own incentive function by accounting for its impact on the learning of the recipients, and through them, the impact on its own extrinsic objective. The task policy is learned via RL. In experiments, agents seem to divide into selfish agents (“winners”) and selfless agents (“cooperators”). Cooperators seem to learn not to send incentives to winners to avoid the cost of sending an incentive. UPDATE AFTER AUTHOR RESPONSE: I have read the author response and think that it sufficiently clarifies some of the questions I had. I am happy to keep my score as is.

Strengths: + The ability to directly influence other agents via reward-giving is a novel improvement to opponent shaping algorithms like LOLA and SOS. I suspect that this paper will be cited by many multiagent learning works in the future. + The paper is clearly written

Weaknesses: - It would be helpful if the paper’s definition of “decentralized” is more explicitly stated in the paper, instead of in a footnote. Other ways of defining “decentralized” is where agents do not have access to the global state and actions of other agents during both training and execution which LIO seems to do. - Systematically studying the impact of the cost of incentivization on performance would have been a helpful analysis (e.g., for various values of \alpha, what are the reward incentives each agent receives, and what is the collective return?). It seems like roles between “winners” and “cooperators” emerge because the cost to reward the other agent becomes high for the cooperators. If this cost were lower, it seems like roles would be less distinguished, causing the collective return to be much lower. - In Figure 5d, more explanation as to why the Winner receives about the same incentive as the Cooperator to pull the lever would be helpful; it doesn’t match how the plot is described on lines 286-287.

Correctness: Yes

Clarity: Yes

Relation to Prior Work: This is perhaps the most relevant paper I have seen (and it has not been cited): Inducing Cooperation through Reward Reshaping based on Peer Evaluations in Deep Multi-Agent Reinforcement Learning by Hostallero et al. AAMAS 2020. Since this paper is recent, I do not think it would be fair to expect this work to be one of the baselines in the experiments. But I do think it is worth discussing how the current work is different from Hostallero et al. in the Related Works section.

Reproducibility: Yes

Additional Feedback:

[Author Response · NeurIPS 2020]

**Reviewer 1.** We appreciate R1's recognition of the novelty of our contribution to MARL and the potential impact on a range of problems. We address R1's two concerns below. **Regarding our chosen baselines**, we note that baselines we include represent three major existing categories: 1) policy gradient and actor-critic with discrete or continuous "give-reward" actions are direct applications of conventional RL (which have been applied to multi-agent incentivization in recent work (Lupu et al. 2020)); 2) LOLA is an archetype of second-order approaches; 3) Inequity Aversion (IA) draws domain knowledge from models in evolutionary biology and sociology to alter individual rewards. Hence we believe the existing baselines allow for fair benchmarking of our new approach. While Social Influence (Jaques et al. 2019) does not have open-source code, we can make an indirect comparison by noting that IA reaches a score around 250 by $1.6 \times 10^8$ steps (Figure 3a in IA (Hughes et al. 2018)), outperforming Social Influence that reaches score of 200 by $3 \times 10^8$ steps (Figure 1a in Jaques et al. 2019) in the original Cleanup map with 5 agents. Hence, the fact that LIO outperforms IA in our experiments implies that LIO compares favorably with Social Influence.

We clarify that LIO technically does not involve a second-order gradient, as the gradients are w.r.t. separate parameters $\theta$ (policy) and $\eta$ (incentive function). **Regarding scalability**, we clarify that the bi-level optimization does not necessarily imply difficulty in scaling up, because the learning of incentives is conducted in a *pairwise* manner: in equation (7) for a fixed reward-giver agent $i$, each term of the summation corresponds to the pair $(i,j)$ for recipient $j \neq i$. Figure 1 shows that LIO attains the global optimum collective reward of $17 (= 2 \times 10 - 3)$ in the Escape Room game with $N = 5$ agents, out of which $M = 3$ agents are incentivized to cooperate despite penalties of $-1$ each. Scaling up to large populations poses new questions regarding population-level phenomenon (such as social norms) that modulate the impact of incentives; we leave this to future work.

Figure 1: Escape Room ($N = 5, M = 3$)

**Reviewer 2.** We appreciate R2's positive feedback on our quantitative results and we are glad that our behavioral analysis of the learned incentive functions provided insight. We believe this work is a suitable contribution to the NeurIPS community, in addition to the broad area of multi-agent learning, as we tackle the open question of emergent cooperation from decentralized learning charted out in Hughes et al. (NeurIPS 2018) by building on general meta-gradient methods (Xu et al., NeurIPS 2018). In our revision, we will elaborate on the wide range of new research questions generated by our work, including theoretical analysis of dynamically-changing incentive functions, new population-level effects in a scaled up context, and adaptive ways to account for the currency of incentives. Regarding Figure 6b where the agent gives nonzero reward for "fire cleaning beam but miss" after 40k steps, one reason is that the agent's actual partner in training already converged to the behavior of consistently cleaning waste successfully (LIO in Figure 6a), so it may have "forgotten" the difference between successful and unsuccessful usage of the cleaning beam. As demonstrated more clearly in the Escape Room results (e.g. Figures 5b and 5d), this can be avoided by choosing a sufficiently large lower bound on the exploration rate by all agents, so that all agents pose the risk of deviating from cooperative behavior, which forces LIO to maintain correct incentivization.

**Reviewer 3.** We thank R3 for recognizing our contribution to the general class of opponent-shaping algorithms. We address each concern as follows. 1) Our definition of "decentralized" focuses solely on the inability to optimize social welfare directly, which is the crux of social dilemmas, and which holds regardless of access to the global state (e.g., Prisoner's Dilemma is fully observable). Hence our definition does not mention the degree of observability. 2) We explain in Appendix A that the coefficient $\alpha$ in the cost for incentivization is indeed an important hyperparameter and provide intuition for how to choose it in practice. We agree that a sweep over $\alpha$ can provide more insight. More broadly, we believe there is room to develop an adaptive scheme to trade off between small $\alpha$, which allows time to learn the effect of incentives, and large $\alpha$, which penalizes redundant incentivization. 3) In Figure 5d, by episode 50k, the cooperator still receives nonzero incentives between 0.5 and 1, but the winner's received incentives has noticeably converged to zero. 4) We became aware of Hostallero et al. (AAMAS 2020) after submission, and we believe there is a crucial methodological difference from our work. They use the temporal difference error of agent $k$'s $Q$-function to modify the rewards of $k$'s peers, so that those peer agents have incentive to take actions that lead to more favorable reward than average for agent $k$. This is a *passive* approach from agent $k$'s viewpoint, and is closer to the method in Hughes et al. (2018), since each agent's original reward is modified by a hand-designed function based on other agents' performance. In contrast, a LIO agent *actively* differentiates through the recipient's learning step to update the learned incentive function, which it uses to change the recipient's total reward. We will include this in the revision.

Hostallero, D. *et al.* (2020). Inducing Cooperation through Reward Reshaping based on Peer Evaluations in Deep Multi-Agent Reinforcement Learning. *AAMAS*.

Hughes, E. *et al.* (2018). Inequity aversion improves cooperation in intertemporal social dilemmas. *NeurIPS*.

Jaques, N. *et al.* (2019). Social influence as intrinsic motivation for multi-agent deep reinforcement learning. *ICML*.

Lupu, A. and Precup, D. (2020). Gifting in multi-agent reinforcement learning. *AAMAS*.

Xu, Z. *et al.* (2018). Meta-gradient reinforcement learning. *NeurIPS*.


[Meta-Review · NeurIPS 2020]

The reviewers are in consensus that this paper provides a useful new framework for sharing rewards in multi-agent RL, along with an algorithm for learning to do so. Some concerns about clarity and the empirical evaluation were resolved via the authors' rebuttal. Hence, the reviewers agree the paper should be accepted.